# Phthalate Exposure Pattern in Breast Milk within a Six-Month Postpartum Time in Southern Taiwan

**DOI:** 10.3390/ijerph18115726

**Published:** 2021-05-26

**Authors:** Shen-Che Hung, Ting-I Lin, Jau-Ling Suen, Hsien-Kuan Liu, Pei-Ling Wu, Chien-Yi Wu, Yu-Chen S. H. Yang, San-Nan Yang, Yung-Ning Yang

**Affiliations:** 1Department of Pediatrics, E-DA Hospital, Kaohsiung 82445, Taiwan; yoshiki6455@gmail.com (S.-C.H.); firecrest31@gmail.com (T.-I.L.); jonathan730608@yahoo.com.tw (H.-K.L.); Peiling0420@gmail.com (P.-L.W.); wucyi1228@yahoo.com.tw (C.-Y.W.); y520729@gmail.com (S.-N.Y.); 2School of Medicine, I-Shou University, Kaohsiung 84001, Taiwan; 3Graduate Institute of Medicine, College of Medicine, Kaohsiung Medical University, Kaohsiung 80708, Taiwan; jlsuen@cc.kmu.edu.tw; 4Research Center for Environmental Medicine, Kaohsiung Medical University, Kaohsiung 80708, Taiwan; 5Department of Medical Research, Kaohsiung Medical University Hospital, Kaohsiung 80708, Taiwan; 6Joint Biobank, Office of Human Research, Taipei Medical University, Taipei 11031, Taiwan; can_0131@tmu.edu.tw

**Keywords:** environmental endocrine disruptor, breast milk, di-(2-ethylhexyl) phthalate

## Abstract

Di-(2-ethylhexyl) phthalate (DEHP), a common plasticizer, has been detected in breast milk in many countries; however, whether phthalate metabolite concentration and the detection rate in breast milk change postpartum is still unknown. We measured phthalate metabolite concentrations in breast milk in the first 6 months postpartum in women enrolled in the E-Da hospital from January to July 2017. A total of 56 breastfeeding mothers and 66 samples were included in this study. We analyzed the samples’ concentration of eight phthalate metabolites using liquid chromatography mass spectrometry. The concentration of mono-2-ethylhexyl phthalate (MEHP) was significantly higher in the first month, and then decreased over time. The detection rate of ono-isobutyl phthalate (MiBP) and mono-n-butyl phthalate (MBP) was low in the first month and then increased over time. Compared with a previous study published in 2011, the levels of MEHP and MiBP in breast milk were much lower in the present study, suggesting an increased awareness of the health risks of phthalate exposure after a food scandal occurred in Taiwan. This study provides information for evaluating newborns’ exposure to different kinds of phthalate through human milk in the postpartum period.

## 1. Introduction

Phthalates (diesters of 1,2-benzenedicarboxylic acid) are used in the synthesis of polymers and as plasticizers in polyvinylchloride (PVC), which are widely employed as industrial chemicals [1]. Many of our daily necessities or consumer products are fabricated from PVC, including packages for food or water [2], plastic bags, toys [3], medical devices [4], lubricants, insect repellents, and personal care products [5,6]; as a consequence, phthalates are widely distributed in food, water, soil, indoor air, and dust [7,8]. In recent years, the production of phthalates has reached about 5 million tons per year [9,10], and in industrial countries around the world, including Taiwan, it is almost impossible to avoid exposure to plastic or plastic-containing products in daily life.

After human exposure, phthalate esters can be rapidly excreted in around 24–48 h [11]; however, they and their metabolites may affect human health and children’s neurological, endocrinal, growth, and reproductive development [12]. According to previous studies, prenatal exposure to phthalates such as di-(2-ethylhexyl) phthalate (DEHP) or di-n-butyl phthalate (DnBP) has been associated with decreased IQ scores in children [13], adverse mental and motor development [14,15], as well as disruption of sex hormone balance in humans [16], with general phthalate exposure being associated with reproductive developmental damage [17,18,19]. In physiological conditions, the decrease in anogenital distance in female infants is associated with the mono-n-butyl phthalate (MBP) found in amniotic fluid [20]. In addition, exposure of male infants to phthalates such as monoethyl phthalate, MBP, mono-benzyl phthalate (MBzP), and mono-isobutyl phthalate (MiBP) is reflected in decreases in anogenital distance [21], sperm quality, and sperm motility [22,23]. For girls, phthalate exposure is associated with altered physiological development of pubic hair and breasts, and menarche time [24]. Moreover, exposure was reported to influence body mass index, inducing increased waist circumference, and contributing to insulin resistance [25]. Due to the potential impact of postnatal phthalate exposure on the associated health risks, the level and composition of phthalate metabolites in human breast milk is worthy to investigate in detail.

Phthalate esters enter the human body in many different ways. Recent studies suggested that infants treated in an intensive care unit might be exposed to DEHP due to the usage of plastic medical devices such as intravenous fluid supply and nasopharyngeal and endotracheal tubes [26]. This iatrogenic exposure to DEHP in an intensive care unit was associated with attention deficit disorder observed in children 4 years after discharge [27]. Even heathy newborns can be exposed to phthalates, as human milk provides the major nourishment for newborns and several studies have reported phthalates and their metabolites in both breast milk and infant formula [28]. Several kinds of phthalate metabolites, such as MEHP, the metabolites of DEHP, monoethyl phthalate, the metabolites of DEP, MnBP, and the metabolites of DBP, have been detected in human breast milk [29]. 

In May 2011, a phthalate food scandal occurred in Taiwan. The Taiwan Food and Drug Administration discovered that DEHP and di-isononyl phthalate (DINP) were being illegally used in more than 900 foodstuffs, including sport drinks, jelly tea drinks, concentrated juice beverages, and food supplements in capsular or powder form [30,31]. Due to costs, two perfumery-chemical companies had intentionally used phthalates instead of palm oil in clouding agents, a key additive in beverages for over 15 years [30]. It was reported that the average daily intake of DEHP was around 0.14 mg/kg bw/day if a person consumed one bottle (500 mL) of sport drink before the food scandal in Taiwan. This estimated daily intake of DEHP is much higher than the government-recommended limit (0.02–0.06 mg/kg body weight/day) [31]. Thus, this incident raised concerns regarding any potential adverse health effects following this illegal addition of phthalates to foodstuffs over the past two decades in Taiwan. The studies published from Taiwan before the food scandal showed that phthalate exposure is associated with altered thyroid hormones in pregnant women, anti-androgenic effects on the fetus, early puberty in girls, sex steroid hormone changes in newborns, and altered motility and chromatin DNA integrity of sperm in men [20,26,32,33,34,35]. Although it is unclear if these associated health risks were posed due to the illegal use of DEHP and DiNP in foods and beverages, Taiwanese people became aware of and avoided this environmental exposure to phthalates [36]. The Taiwan government also reclassified these phthalates from class IV to class I toxic chemicals, destroyed all the contaminated food products, and opened consultation clinics in hospitals island-wide for health screening [37]. However, whether the levels of phthalate metabolites in human milk have decreased due to the awareness of food safety issues in recent years remains a concern.

Human milk is a complex matrix. Colostrum (also known as beestings, bisnings (the obsolete form), or first milk) is the first form of milk produced by the mammary glands immediately following delivery of the newborn, which then gradually changes to mature milk over the course of several weeks. Colostrum contains higher protein levels, including antibodies, complement, and cytokines, but lower fat [38,39]. Most studies focused on the determination of phthalate diesters, monoesters, and other metabolites in human mature milk; however, few have focused on phthalate metabolite detection in colostrum. Whether DEHP metabolites are present in colostrum and whether DEHP metabolite concentrations vary in different periods of human milk are still unknown.

In this study, the DEHP metabolite concentrations in breast milk were measured and the concentrations compared over different time periods during the first 6 months after delivery, and then compared concentrations reported in previous studies to elucidate the change in phthalate exposure in breast milk in a lactating postpartum Taiwanese population. The results provide further understanding of how newborns are exposed to phthalates in breast milk and suggest a pathway to interventions that will avoid such neonate exposure in the future.

## 2. Materials and Methods

### 2.1. Study Design and Sample Collection

Breast milk from mothers aged between 20 and 40 years was collected at E-Da Hospital in Kaohsiung, Southern Taiwan. The protocol of this study was approved by the Institutional Review Board of E-Da Hospital (EMRP47104N), and the study was conducted in accordance with the ethical standards set out in the Declaration of Helsinki. Lactating women in the outpatient department or the postpartum care center were invited to participate in our study. All the samples were collected from the first week up to 6 months after delivery. Sixty-six samples were collected from January to April 2017, with 47 mothers contributing breast milk once, 8 mothers contributing twice, and 1 mother contributing three times. When a mother collected breast milk more than once, the duration between collected times was at least one week. 

The demographic information is provided in Table 1, including age of the mother, gestational age at delivery, education level, parity, delivery type, and sex of the baby. We also recorded whether the mothers ate traditional meals during the one-month period of confinement after childbirth. Confinement is a special custom in Chinese culture. During the confinement period, the mother does not go outside and should eat nutritious food prepared using special cooking methods, including stewed chicken, fried eggs, and fish, which are cooked with sesame oil and rice wine [40]. 

### 2.2. Instrument Settings

Phthalate metabolites in human breast milk were measured using a Waters ACQUITY UPLC system (Waters Corporation, Milford, MA, USA) coupled with tandem MS (Finnigan TSQ Quantum Ultra triple-quadrupole MS, Thermo Electron, San Jose, CA, USA) in combination with Xcalibur software (ThermoFinnigan, Bellefonte, PA, USA). The LC–MS–MS system was equipped with an electrospray ion source (ESI) and was run in negative mode. The injection volume was 10 μL on an ACQUITY UPLC CSH Phenyl-Hexyl Column (130 Å, 1.7 µm, 2.1 × 100 mm, Waters Corporation, Milford, MA, USA) equipped with a filter (Waters Acquity UPLC™ BEH C18 column, 1.7 μm, 2.1 × 5 mm) in front of the column. The flow rate was 250 μL/min, and the column temperature was 40 °C. Solvents were A: 0.1% acetic acid in water and B: 0.1% acetic acid in acetonitrile. Solvent programming was 0.0–5.0 min, 30% B; 10.0 min, 40% B; 12.0 min, 50% B; 13.0 min, 100% B; and 13.1–15.0 min, 10% B. The limit of detection (LOD), limit of quantitation (LOQ), and linear range of standard curves for phthalate metabolites are summarized in Appendix A. The analyzed phthalate metabolites included mono-2-ethylhexyl phthalate (MEHP), mono-(2-ethyl-5-carboxypentyl) phthalate (MECPP), mono-(2-ethyl-5-hydroxyhexyl) phthalate (MEHHP), mono-(2-ethyl-5-oxohexyl) phthalate (MEOHP), mono[2-(carboxymethyl)hexyl] phthalate (MCMHP), MBzP, MiBP, and MBP.

### 2.3. Sample Collection and Preparation

The total amount of human milk was collected from both breasts using an electric pump into a sterile glass bottle. After gently mixing, around 10 mL of breast milk sample was aliquoted into a 50 mL tube (CT-50-PL-TW, Protech, Taiwan), and then stored at −20 °C until analysis. A 1 mL of human breast milk sample was thawed; centrifuged to collect its supernatant; transferred to a glass tube; added to various ^13^C_4_-labeled internal standards such as MEHP, MECPP, MEHHP, MEOHP, MCMHP, MBzP, MiBP, and MBP; and then buffered with ammonium acetate and beta-glucuronidase/aryl-sulfatase. The samples were incubated in a 37 °C water bath for 90 min, allowed to sit at room temperature for 10 min, and then acidified by phosphate buffer, vortex-mixed, and centrifuged at 3500 rpm for 10 min. The supernatant was loaded into a solid-phase extraction cartridge (Strata XL, Phenomenex, Allerød, Denmark; 200 mg, 3 mL). We sequentially loaded 1% phosphoric acid and 10% acetonitrile for washing. Acetonitrile (ACN) and ethyl acetate (EA) were added to elute metabolites, and then dried with nitrogen gas at 55 °C. Finally, the residues were reconstituted with 1% formic acid in 50% methanol/water, and subjected to LC–MS–MS.

### 2.4. Statistical Analysis

Metabolite levels under the LOD were calculated as half the LOD value, whereas the values of continuous variables were compared using a Mann–Whitney U-test for two-group comparison or a Kruskal–Wallis test for three-group comparison. The detection rates of phthalate metabolites were compared using Fisher’s exact test. Data were managed and analyzed with commercially available software (SPSS, version 24, IBM Corp., Armonk, NY, USA), with a *p* value of 0.05 set to define statistical significance. Linear regression was used to determine the correlation between the time and concentrations of DEHP metabolites.

## 3. Results

### 3.1. General Characteristics of the Study Subjects

A total of 56 breastfeeding mothers and 66 samples were included in this study. The characteristics of the participants are shown in Table 1. Among these study subjects, the mean age of the mothers was 32.5 years, and more than 80% of the subjects had university-level or higher education. The mean gestational age at delivery was 269.9 days, and all expectant women had a gestational age greater than 36 weeks.

### 3.2. Breast Milk Phthalate Metabolites Concentrations and Detection Rates

A previous study detected parental compounds DEP, di-n-butyl phthalate (DBP) and DEHP in human breast milk [41]. Here, we evaluated metabolites including MEHP, MECPP, MEHHP, MEOHP, MCMHP, MBzP, MiBP, and MBP in human breast milk, and the concentrations of phthalate metabolites and detection rates in breast milk are shown in Table 2, with MEHP, the primary metabolite of DEHP, having the highest detection rate (87.88%) with a median concentration of 0.3 μg/L. The other secondary and tertiary metabolites of DEHP (MCMHP, MEHHP, MEOHP, and MCMHP) were detected as 37.88%, 40.91%, 30.30%, and 28.79%, respectively. MiBP and MBP, the representative metabolites from di-isobutyl phthalate (DiBP) and DBP, respectively, also had >50% detection rates in human milk; the median concentrations of MiBP and MBP were 0.3 and 0.5 μg/L, respectively. In contrast with all DEHP metabolites and MBzP, the detection rates of MiBP and MBP were the lowest during the first month and highest three months later.

### 3.3. Sources of Phthalate Exposure

Breast milk is the only natural nutrition resource of newborns, and as the safety of human milk should be warranted, the demographic data of mothers were analyzed to determine any association with phthalate metabolites in breast milk. The demographic characteristics, including education level of the mother, parity, and sex of the baby revealed no correlation with MEHP concentration, although interestingly, it was higher in the confinement meal group versus the no-confinement meal group (*p* = 0.007; Table 3).

### 3.4. Variation in DEHP Metabolite Levels in Different Periods

The results showed that the average MEHP level was 0.05–2.1 μg/L in first-month breast milk, 0.05–1.9 μg/L in 1~3-month samples, and 0.05–0.5 μg/L after 3 months in breast milk samples (Table 4). We found that the MEHP concentration of breast milk during the first month postpartum was significantly higher than during the other two time periods. We further observed a negative correlation between MEHP level in breast milk and postpartum period (Figure 1).

We further compared our results with those of another Taiwan study published by Lin et al. in 2011 [42], which was performed before the phthalate incident was discovered. As shown in Table 5, we found that the median MEHP level, the primary metabolite of DEHP, has dramatically decreased (median: 3.6 vs. 0.3 μg/L) in breast milk, whereas the levels of MiBP remained similar, and MBzP was undetectable as reported in both studies. In addition, the highest level of MEHP in the 2011 study was 46.53 μg/L, being significantly higher than the highest level of 2.1 μg/L found in the present study; MiBP highest level was found to be higher in the previous cohort than that in the present cohort.

## 4. Discussion

Humans can be easily exposed to DEHP, a ubiquitous environmental endocrine disruptor, through food, inhalation, and skin [29,36]. Studies revealed that DEHP metabolites are detectable in human breast milk in various countries [29]. In this study, the metabolites of phthalate were detectable in breast milk in South Taiwan in 2017. MEHP was the most detected metabolite in human breast milk, suggesting postpartum mothers in this area are mainly exposed to this phthalate; interestingly, detectable DEHP metabolites in breast milk appears to be associated with the meal style during the postpartum period. This potential exposure source of DEHP requires further detailed investigation.

DEHP is applied globally in a variety of daily-life products [30]. Due to their ubiquitous distribution, phthalate metabolites are detectable in women and human breast milk in various countries including Korea and in Europe [29]. In 2011, Lin et al. [42] reported the level of DEHP metabolites in breast milk in Taiwan; however, the levels of DEHP metabolites were relatively low in human milk in the present study. It is possible that people in Taiwan have become more aware about avoiding phthalate exposure through the purchase and intake of food products, and illegal phthalate addition in foods has been found and prohibited by the government following the Taiwan food scandal in 2011 [30,43].

In addition to DEHP exposure, it was observed that the mothers might have been exposed to DiBP and DBP 1 month after delivery. DiBP exposure is reported as being higher through hair spray use and makeup products [44]; urine MBP levels were found to be higher in those drinking beverages from plastic cups and young women with a low frequency of handwashing [45]. It is expected that mothers need considerable time to care for their baby during the first month after delivery, and the traditional first-month period of “doing the month” in Taiwanese culture might limit DiBP and DBP exposure after delivery due to less application of makeup and a higher frequency of handwashing. After they return to work, exposure levels to DiBP and DBP might increase, although the sources of DiBP and DBP need to be investigated in detail.

Our findings suggest that DEHP exposure might have decreased in pregnant women and postpartum women over the past decade; however, a negative correlation was observed between MEHP levels in breast milk and the postpartum period. The reason for this negative correlation might be the different lipid profiles in colostrum and mature milk [39]. However, as colostrum has a lower lipid content than mature milk [39,46], the level of MEHP, which is hydrophobic, would be expected to be lower in colostrum than in mature milk [41]. Therefore, the distinct lipid profile in breast milk cannot explain the finding in the present study. The other possible explanation is that the confinement meals for postpartum women might be a potential DEHP source [47] due to the traditional and unique food culture for postpartum women in Asian countries. During this postpartum period (usually one month in Taiwan), immediately after delivery, women face restrictions on their diet of “confinement meals” [40]. Most of these meals are packaged with plastic materials and microwaved with PVC-made plastic wrap [48], and are usually composed of foods with high lipid content such as stewed chicken, fish soup, and fried eggs. As DEHP is a lipophilic chemical that easily migrates from plastic containers or plastic-wrap covers to meals [45], it is highly possible that DEHP levels are much higher in confinement meals versus a regular diet that a mother would usually eat 1 month after delivery. Therefore, the food safety issue of confinement meals for postpartum women needs to be further investigated as human milk is the best food source for newborns in the first few months of life.

Animal studies showed that early-life exposure to DEHP or its metabolites might generate health concerns later in life [49], suggesting a potential health risk to newborns through the uptake MEHP-contaminated breast milk from mothers. DEHP was shown to have endocrine-disrupting effects on development and reproductive systems in rodents [50], particularly with exposure at the prenatal and perinatal stages [51]. It seems that the fetus is most sensitive to the anti-androgenic effect of phthalates during development [52]. In addition, our recent study showed that chronic exposure to low-dose DEHP in pregnant dams significantly enhanced allergic lung inflammation in young offspring through at least four generations [53]. Taken together, considering the potential MEHP modulatory effect on neonates, the DEHP daily intake of newborns should be considered.

Human tolerable daily intake (TDI) is estimated using the no observed adverse effect level (NOAEL) from animal toxicological studies evaluating the human overdose intake level for diesters such as DEHP. The TDI of DEHP in adults is 50 μg/kg body weight/day as determined by the European Food Safety Authority (EFSA) [54]. The TDI value of DEHP is 20 and 25 μg/kg body weight/day in 0–3-month-old newborns or 3–12-month-old infants, respectively, according to the European Union Risk Assessment Report from European Chemicals Bureau (ECB) [55]. As newborns consume up to 180 mL breast milk/kg body weight during the first week of life [56], the median (range) of MEHP daily intake from colostrum is around 0.063 (0.009–0.378) μg/kg body weight in the present study, suggesting the safety human milk in terms of plasticizer exposure. However, information to estimate the daily intake of all types of phthalate chemicals from breast milk in infants needs to be further investigated [57,58,59].

Due to the phthalate incident in 2011 [43], Taiwanese people are now concerned about whether the body burden of phthalate has decreased since the incident. Thus, we compared our findings with those from the 2011 study concerning phthalate exposure in human milk in Taiwan [42] and found that, although the detection rates in the present study seem higher than those in Lin et al.’s study [42], the levels of MEHP have significantly decreased during these years, suggesting that Taiwanese people have developed increased awareness of the safety of food, and DEHP contamination in foodstuffs appears to have been controlled by the Taiwan government. Accordingly, breast milk appears to be safer now than before in terms of plasticizer contamination. The high detection rates of phthalate metabolites in the present study may be due to the better sensitivity of equipment used nowadays.

This study has some limitations. Firstly, although 56 breastfeeding mothers were recruited, the sample size was still limited. Secondly, it would be informative to analyze continuous milk samples from one postpartum woman due to the short half-life of DEHP in humans [10]; thirdly, the maternal urinary and serum levels of DEHP metabolites were not analyzed in this study as metabolite levels in breast milk were not well-correlated with those in urine or serum [58]. Lastly, many special maternal conditions including obesity or medication may change human milk composition [46]. We did not investigate maternal disease or medication in the present study.

## 5. Conclusions

In Taiwan, the levels of phthalate metabolites in human milk have decreased in recent years. This indicates that people have become more aware of environmental pollutants in foods and the government has well controlled this food-safety issue, although for postpartum mothers, phthalate metabolite concentrations appear high in the first month after delivery but then decrease with time. It would be worthwhile to investigate the potential sources in detail as the potential effects of phthalate metabolites in human milk on the development of newborns and later in life are sufficiently concerning.

## Figures and Tables

**Figure 1 ijerph-18-05726-f001:**
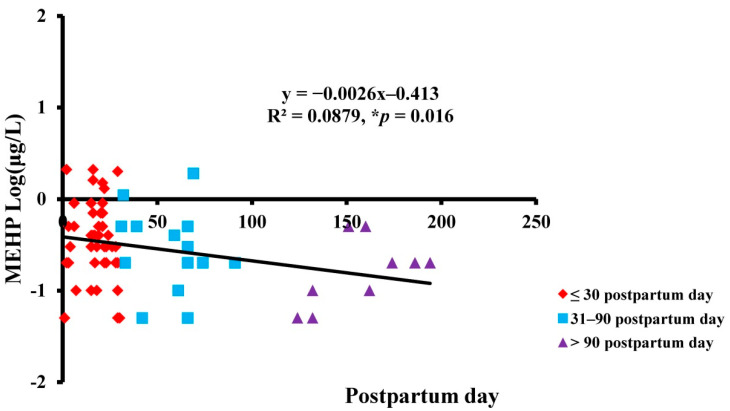
The negative correlation between MEHP level in breast milk and postpartum period. The human milk from different time periods postpartum is labeled with different markers. Red diamonds indicate <30 days postpartum, blue squares indicate 30–90 days postpartum, and purple triangles indicate >90 days postpartum. * *p* < 0.05 is considered significant using linear regression. y = −0.0026x−0.413, R^2^ = 0.0879, and r = −0.016.

**Table 1 ijerph-18-05726-t001:** Characteristics of the study population.

	Mean ± SD (%) or *n* (%)	Median (Range)
**Mother (*n* = 56)**		
Age (years)	32.5 ± 4.9	33.1 (20.4–41.6)
Gestational age at delivery (days)	269.9 ± 9.2	271 (245–289)
**Education level (*n* = 56)**		
High school or lower	11 (19.6)	
University or higher	45 (80.4)	
**Parity (*n* = 56)**		
Primipara	30 (53.6)	
Multipara	26 (46.4)	
**Delivery type (*n* = 56)**		
Vaginal delivery	38 (67.9)	
Caesarean section	18 (32.1)	
**Baby sex (*n* = 56)**		
Male	29 (51.8)	
Female	27 (48.2)	

**Table 2 ijerph-18-05726-t002:** Phthalate metabolites levels and detection rates in breast milk samples.

Parental Phthalate (*n* = 66)					
Corresponding Phthalate Metabolite		Median (μg/L)	Range (μg/L)	Detection Rate (%)	*p* Value (Detection Rate)
**DEHP**					
**MEHP (*n* = 66)**		**0.3**	**0.05–2.1**	**87.88**	
≤30	postpartum day (*n* = 42)	0.35	0.05–2.1	92.86	0.137
31–90	postpartum day (*n* = 14)	0.25	0.05–1.9	85.71	
>90	postpartum day (*n* = 10)	0.15	0.05–0.5	70.00	
**MECPP (*n* = 66)**		**0.05**	**0.05–0.74**	**37.88**	
≤30	postpartum day (*n* = 42)	0.05	0.05–0.74	47.62	0.122
31–90	postpartum day (*n* = 14)	0.05	0.05–0.4	21.43	
>90	postpartum day (*n* = 10)	0.05	0.05–0.4	20.00	
**MEHHP (*n* = 66)**		**0.05**	**0.05–0.3**	**40.91**	
≤30	postpartum day (*n* = 42)	0.05	0.05–0.3	38.10	0.473
31–90	postpartum day (*n* = 14)	0.05	0.05–0.3	35.71	
>90	postpartum day (*n* = 10)	0.2	0.05–0.3	60.00	
**MEOHP (*n* = 66)**		**0.05**	**0.05–0.6**	**30.30**	
≤30	postpartum day (*n* = 42)	0.05	0.05–0.36	26.19	0.116
31–90	postpartum day (*n* = 14)	0.05	0.05–0.3	21.43	
>90	postpartum day (*n* = 10)	0.3	0.05–0.6	60.00	
**MCMHP (*n* = 66)**		**0.1**	**0.1–0.8**	**28.79**	
≤30	postpartum day (*n* = 42)	0.1	0.1–0.41	23.81	0.165
31–90	postpartum day (*n* = 14)	0.25	0.1–0.4	50.00	
>90	postpartum day (*n* = 10)	0.1	0.1–0.8	20.00	
**BBzP**					
**MBzP** **(*n* = 66)**		**0.1**	**0.1–0.9**	**27.27**	
≤30	postpartum day (*n* = 42)	0.1	0.1–0.88	26.19	0.721
31–90	postpartum day (*n* = 14)	0.1	0.1–0.9	35.71	
>90	postpartum day (*n* = 10)	0.1	0.1–0.8	20.00	
**DiBP**					
**MiBP (*n* = 66)**		**0.3**	**0.2–3.3**	**54.55**	
≤30	postpartum day (*n* = 42)	0.2	0.2–3.3	42.86	0.047 *
31–90	postpartum day (*n* = 14)	0.3	0.2–0.9	71.43	
>90	postpartum day (*n* = 10)	0.3	0.2–0.6	80.00	
**DBP**					
**MBP (*n* = 66)**		**0.5**	**0.2–5.9**	**60.61**	
≤30	postpartum day (*n* = 42)	0.2	0.2–5.9	45.24	0.002 *
31–90	postpartum day (*n* = 14)	0.6	0.2–3.2	92.86	
>90	postpartum day (*n* = 10)	0.6	0.2–1.0	80.00	

* *p* < 0.05 per Fisher’s exact test. The data in bold represents results from all samples (*n* = 66). The gray is used to identity DEHP, BBzP, DiBP, and DBP as parental phthalates. All these terms are in bold type and gray area.

**Table 3 ijerph-18-05726-t003:** Demographic characteristics and MEHP level in breast milk of study subjects.

	MEHP (μg/L) Mean ± SD (%)	*p* Value (Mann–Whitney U)
**Education level (mother)**		
High school or lower (*n* = 11)	0.595 ± 0.671	0.450
University or higher (*n* = 55)	0.455 ± 0.474	
**Parity (mother)**		
Primipara (*n* = 39)	0.444 ± 0.443	0.398
Multipara (*n* = 27)	0.530 ± 0.600	
**Food (mother)**		
Confinement meal (*n* = 39)	0.551 ± 0.514	0.007 **
No confinement meal (*n* = 27)	0.374 ± 0.498	
**Sex (baby)**		
Male (*n* = 32)	0.475 ± 0.541	0.314
Female (*n* = 34)	0.482 ± 0.489	

** *p* < 0.01.

**Table 4 ijerph-18-05726-t004:** MEHP levels in breast milk during different postpartum periods.

MEHP		Median (Range) (μg/L)	*p* Value (Kruskal–Wallis Test)
≤30	postpartum day (*n* = 42)	0.35 (0.05–2.1)	* 0.038
31–90	postpartum day (*n* = 14)	0.25 (0.05–1.9)	
>90	postpartum day (*n* = 10)	0.15 (0.05–0.5) *	

* *p* < 0.05.

**Table 5 ijerph-18-05726-t005:** Comparison of phthalate metabolite levels in human breast milk.

	2011 [42]		The Present Study	
Phthalate metabolite	Median (range) (μg/L)	Detection rate (%)	Median (range) (μg/L)	Detection rate (%)
MEHP	3.6 (<0.25–46.53)	73.33	0.3 (0.05–2.1)	87.88
MiBP	0.5 (<0.5–39.70)	33.33	0.3 (0.2–3.3)	54.55
MBzP	<0.25 (<0.25–0.7)	10.00	0.1 (0.1–0.9)	27.27

## Data Availability

The datasets used and/or analyzed during the current study are available from the corresponding author upon reasonable request.

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
