# Peer review of "Phthalate Exposure Pattern in Breast Milk within a Six-Month Postpartum Time in Southern Taiwan"

_ijerph, 2021, doi:10.3390/ijerph18115726_

Round 1
Reviewer 1 Report
I have read this paper with great interest, and highly value the observations as reported. My comments are provided sequential, but the bio-analysis and the collection strategy are the ‘major’ relevant comments.
Textual suggestion: line 51-52 on pg 2
Why do you add ‘even’ in the text line on amniotic fluid ?
Phthalate in nicu/picu: suggest to add the Verstraete et al, Intensive Care Med 2016 paper, describing an association between circulating phthalates during critical illness and long-term attention deficit, as supportive to the message and concept of the paper.
The final exposure to the newborn or infant is also driven by the ingested volumes, as reflected in the paper of Yeung et al (Matern Child Nutr 2020) this is of relevance for the colostrum, and the first days of postnatal life to reach 150-180 ml/kg at the end of the first week of life.
The collection strategy should be better described. Where ‘full volumes’ collected, or samples at the end of a breastfeeding session, etc ? perhaps (somewhat out of scope), you can consider the guidelines of the FDA on lactation studies for some inspiration, but in my opinion, the most relevant is a detailed description of the procedure as used in the current study (https://www.fda.gov/regulatory-information/search-fda-guidance-documents/clinical-lactation-studies-considerations-study-design ).Where the colostrum samples collected, or not ?
Bio-analysis: any data on LLQ versus detection limits ? range of linearity ?
Can you explain ‘confinement’ meal, as explored as one of the covariates of MEHP, and has been subsequently discussed in the discussion section. This need a description in the methods section of the paper.
Figure 1: suggest to add the r and p value to the legend.
Can the authors somewhat further elaborate on this negative correlation ? does this relate to changes in breast milk composition ? we refer to a recent paper on approaches to study ‘drug’ or polluants into breast milk (Nauwelaerts et al, BioMed Pharmacother 2021) for mechanistic patterns
Reviewer 2 Report
The authors have analyzed phthalate contamination in breast milk. The rationale is based on a food safety incidence in Taiwan which has occured in 2011. It is therefore of high importance to study the level of contamination of breast milk by phthalates because they are endocrine disruptors. It means that they are deleterious at low doses and that infants are vulnerable populations with regards to the DOHAD hypothesis.
However, I have several comments which taken into consideration should improve the interest of the paper.
The Introduction section should give informations on the incident; levels of contamination; which phthalate; what consequences in terms of risk assessment policies. Next, describe what are the consequences of phthalate exposure. The anogenital distance is a good parameter for boys because reduced anogenital distance is a marker of feminization but for girls? Also indicate if half-lives are different among phthalates? do they store in fat that will explain accumulation in milk. In addition, give explanations for higher levels in colostrum;
Results should be results. It means that they do not have to be discussed in the Results section. It is confusing.
In the discussion section, discuss data without mixing everything
Author Response
Reviewer 2
The authors have analyzed phthalate contamination in breast milk. The rationale is based on a food safety incidence in Taiwan which has occurred in 2011. It is therefore of high importance to study the level of contamination of breast milk by phthalates because they are endocrine disruptors. It means that they are deleterious at low doses and that infants are vulnerable populations with regards to the DOHAD hypothesis.
However, I have several comments which taken into consideration should improve the interest of the paper.
Q1: The Introduction section should give informations on the incident; (a) levels of contamination; which phthalate; what consequences in terms of risk assessment policies. Next, describe what are the consequences of phthalate exposure. (b) The anogenital distance is a good parameter for boys because reduced anogenital distance is a marker of feminization but for girls? (c) Also indicate if half-lives are different among phthalates? (d) do they store in fat that will explain accumulation in milk. (e) In addition, give explanations for higher levels in colostrum.
RESPONSE:
Q1(a) Regarding the incident, we have added a new paragraph in the Introduction section (lines 73–92) in the revised manuscript. It is shown below.
“In May 2011, phthalate food scandal happened in Taiwan. The Taiwan Food and Drug Administration discovered that DEHP and di-isononyl phthalate (DINP) were il-legally used in more than 900 foodstuffs, including sport drinks, jelly tea drinks, con-centrated juice beverages, and food supplements in capsular or powder form [30, 31]. If one person drinking one bottle of sport drinks (around 500 ml) would expose to DEHP about 0.14 mg/kg body weight/day, which is much higher than government guidelines (0.02–0.06 mg/kg body weight/day) [31]. Thus, this incident raises the concerns regard-ing any potential adverse health effects following this illegal addition of phthalates in foodstuffs over the past two decades in Taiwan. The studies published from Taiwan before the food scandal happened showed the phthalate exposure were associated with altered thyroid hormones in pregnant women, anti-androgenic effects on the fetus, early puberty in girls, sex steroid hormone change in newborns, altered motility and chromatin DNA integrity of sperm in men [20, 26, 32-35]. Although it is not certain that these associated health risks are due to the illegal use of DEHP and DiNP in foods and beverages, Taiwanese people became aware of and avoided this environmental expo-sure to phthalates [36]. Taiwan government also reclassified these phthalates from class IV to class I toxic chemicals, destroyed all the contaminated food products, and opened the consultation clinics in hospitals island-wide for health screening [37]. However, it remains a concern as to whether the levels of phthalate metabolites in human milk have decreased due to awareness of food safety issues during recent years.”
Q1(b) For girls, phthalate exposure is associated with altered physiological development of pubic-hair development, breast development, and menarche time [24]. This information has been added in the Introduction section in the revised manuscript (lines 56–58).
Q1(c) The half-life values of all phthalate chemicals detected in the present study are slightly different; however, all of them are less than 24 hours. Please refer to the reference 11 cited in the revised manuscript (lines 44–45).
Q1(d) Compared to urine, human breast milk contains higher level of hydrophobic phthalate monoester metabolites and less level of hydrophilic secondary or tertiary metabolites. This is because breast milk contains high lipid content compared to urine. This differential distribution is particularly concern as the hydrophobic phthalates (such as DEHP and MEHP) are shown to have adverse effects following transmaternal exposure through in utero or lactation in rodent studies. Please refer to reference 41 in the revised manuscript.
Q1(e) Regarding the reason for this negative correlation, one might expect that it is because of the different lipid profile in colostrum and mature milk [39]. However, as colostrum has lower lipid content than mature milk [39] [46], it suggests that the level of MEHP with hydrophobic property would be expected lower in colostrum than in mature milk [41]. Therefore, the distinct lipid profile in breast milk cannot explain the finding in the present study. The other possible explanation is that the confinement meals for post-partum women might be the potential DEHP source [47] due to the traditional and unique food culture for postpartum women in Asian countries. This information has been added to the Discussion section (lines 257–265) in the revised manuscript.
2、Results should be results. It means that they do not have to be discussed in the Results section. It is confusing.
RESPONSE:
Thanks for the critical suggestion. Please refer to the red mark in the Result section in the revised manuscript.
3、In the discussion section, discuss data without mixing everything
RESPONSE:
We have revised the content in the Discussion section for a clear presentation. Please refer to the red mark in the Discussion section in the revised manuscript.

Round 2
Reviewer 2 Report
The ms has been improved. Some comments:
rewrite lines 51-56 (cut in 2 sentences); lines 66-67; line 76-77
I did not reada throughout the ms why industrials added phtalates in foodstuffs. I think this is important to understand the reason.
line 269: it is saif that DEHP is lipophilic. yet line 309 that it is a short-life chemical which is true. Could the authors indicate references mentionning that DEHP is lipophilic and moderate the affirmation as it is known that DEHP has a very short half-life because it is rapidly metabolized. Lipophilic chemicals are hardly metabolized
line 317: certainly people in Taiwan are careful at not being contaminated with phthalates. But it should also be indicated that authorities must keep on eye, as well.
